# How Has the Aspergillosis Case Fatality Rate Changed over the Last Two Decades in Spain?

**DOI:** 10.3390/jof8060576

**Published:** 2022-05-27

**Authors:** Pablo González-García, Montserrat Alonso-Sardón, Beatriz Rodríguez-Alonso, Hugo Almeida, Ángela Romero-Alegría, Víctor-José Vega-Rodríguez, Amparo López-Bernús, Juan Luis Muñoz-Bellido, Antonio Muro, Javier Pardo-Lledías, Moncef Belhassen-García

**Affiliations:** 1Servicio de Medicina Interna, Hospital Marqués de Valdecilla, IDIVAL, Universidad de Cantabria, 39008 Santander, Spain; pablo.gonzalezg@scsalud.es (P.G.-G.); javipard2@hotmail.com (J.P.-L.); 2Área de Medicina Preventiva, Epidemiología y Salud Pública, IBSAL, CIETUS, Universidad de Salamanca, 37007 Salamanca, Spain; sardonm@usal.es; 3Servicio de Medicina Interna, Unidad de Enfermedades Infecciosas CAUSA, IBSAL, CIETUS, 37007 Salamanca, Spain; beamedicina@gmail.com (B.R.-A.); aralegria@yahoo.es (Á.R.-A.); alopezb@saludcastillayleon.es (A.L.-B.); 4Serviçio de Medicina Interna, Unidade Local de Saúde de Guarda, 6300 Guarda, Portugal; hugoalmeida6@gmail.com; 5Servicio de Medicina Interna, CAUSA, 37007 Salamanca, Spain; vjvega92@gmail.com; 6Servicio de Microbiología y Parasitología, CAUSA, CIETUS, IBSAL, Departamento de Ciencias Biomédicas y del Diagnóstico, Universidad de Salamanca, CSIC, 37007 Salamanca, Spain; jlmubel@usal.es; 7Infectious and Tropical Diseases Group (e-INTRO), IBSAL-CIETUS, Faculty of Pharmacy, University of Salamanca, 37007 Salamanca, Spain; ama@usal.es

**Keywords:** aspergillosis, human aspergillosis, invasive fungal diseases, case fatality rate, Spain

## Abstract

(1) Background: *Aspergillus* produces high morbidity and mortality, especially in at-risk populations. In Spain, the evolution of mortality in recent years due to this fungus is not well established. The aim of this study was to estimate the case fatality rate of aspergillosis in inpatients from 1997 to 2017 in Spain. (2) Methodology: A retrospective descriptive study was conducted with records of inpatients admitted to the National Health System with a diagnosis of aspergillosis. (3) Principal findings: Of 32,960 aspergillosis inpatients, 24.5% of deaths were registered, and 71% of the patients who died were men. The percentage of deaths increased progressively with age. The case fatality rate progressively decreased over the period, from 25.4 and 27.8% in 1997–1998 to values of 20.6 and 20.8% in 2016 and 2017. Influenza and pneumonia occurrence/association significantly increased case fatality rates in all cases. (4) Conclusions: Our study shows that lethality significantly decreased in the last two decades despite the increase in cases. This highlights the fact that patients with solid and/or hematological cancer do not have a much higher mortality rate than the group of patients with pneumonia or influenza alone, these two factors being the ones that cause the highest CFRs. We also need studies that analyze the causes of mortality to decrease it and studies that evaluate the impact of COVID-19.

## 1. Introduction

Aspergillosis is caused by species of the fungus *Aspergillus* spp., with *A. fumigatus*, *A. flavus*, *A. niger* and *A. terreus* being the species most frequently involved [1]. They are usually saprotrophic fungi, and their primary habitat is the soil. There are a wide variety of lung infections caused by *Aspergillus*, which are determined by the interaction between the host’s immunity and the fungi. Thus, in a state of hyper-immunity *Aspergillus* cause allergic bronchopulmonary aspergillosis. Minor immunological defects or deformities such as stenosis or bronchiectasis are involved in the colonization of the bronchial tree and the chronic pulmonary aspergillosis (chronic necrotizing aspergillosis, aspergilloma). Finally, neutropenic and other severe immunologic disorders are usually involved in invasive aspergillosis (IA) [2]. In addition to the pulmonary forms, *Aspergillus* can cause infection in skin, bone, and other tissues. The HIV/AIDS pandemic, tuberculosis, chronic obstructive pulmonary disease (COPD), asthma, and the increasing incidence of cancers are the major drivers of fungal infections in both developed and developing countries globally [3]. In developed countries, increasing use of immunosuppressors, allogeneic hematopoietic stem cell transplants, solid organ transplants, and other factors could be associated with higher counts of patients infected by this fungus [4,5]. Recently, influenza and SARS-CoV-2 infection were identified as independent risk factors for invasive pulmonary aspergillosis and are associated with high mortality [6,7,8,9,10]. Therefore, fungal infection of the *Aspergillus* spp. genus has become one of the most frequent causes of death in severely immunocompromised patients, with mortality rates reaching up to 50% [3]. Epidemiological studies in the mid-2000s reported a decrease in the incidence and mortality rates of IA in allogeneic HSCT and in SOT recipients. These results could be due to (i) changes in HSCT practices leading to a shorter duration of pre-graft neutropenia and (ii) better strategies for the diagnosis, prevention, and treatment of fungal diseases [11,12,13]. In Spain, two previous works have shown an increase in the count of aspergillosis [14,15]. A few works have shown that overall mortality due to aspergillosis could be decreasing [3,16]. The aim of this study was to estimate the fatality rate in hospitalized patients with aspergillosis in the Spanish National Health System (NHS) during the period from 1997 to 2017 as a measure of disease severity.

## 2. Materials and Methods

### 2.1. Study Population and Data Source

We designed a descriptive longitudinal and retrospective study on patients diagnosed with aspergillosis and hospitalized in public hospitals of the Spanish NHS between 1 January 1997, and 31 December 2017.

The Spanish NHS is defined by universal coverage and a broad portfolio of services that includes all technologies and health procedures for diseases. It is integrated by all the health services of the State Administration and the health services of the 17 Autonomous Communities that integrate up the Spanish state.

Data were obtained from the Minimum Basic Dataset (Conjunto Minimo Basico de Datos, CMBD in Spanish). These data were provided by the Health Information Institute of the Ministry of Health and Equality. CMBD is the main database for information on morbidity and the care process of patients treated in specialized care. It provides demographic data (sex, age, and place of residence), clinical variables (diagnoses and procedures), and variables related to hospital episodes, such as circumstances of admission (urgent or programmed), patient discharge (discharge to home, transfer to another hospital or death), and average length of stay. Diagnoses and procedures were coded using the International Classification of Diseases, 9th edition, Clinical Modification (ICD-9-CM), and 10th edition (ICD-10). Principal diagnosis is defined as the post-study condition that caused the hospital admission, according to the ICD-9-CM or ICD-10 Official Guidelines for Coding and Reporting. Secondary diagnoses (up to 13 in ICD-9-CM/up to 19 in ICD-10) are “other diagnoses” or conditions that coexisted at the time of admission or developed subsequently and affected the patient´s care during the actual episode.

### 2.2. Selection of Aspergillosis Patients

All patients admitted to NHS public hospitals from 1997 to 2017 with a principal and/or secondary diagnosis of aspergillosis according to the International Classification of Disease 9th edition, Clinical Modification (ICD-9-CM), code 117.3, cases 1997–2015; and 10th edition (ICD-10), code B44, cases 2016–2017 were included. Patients with missing data were excluded. Evidence is growing that aspergillosis in the pediatric population has different epidemiology and outcomes than in adults. For this reason, the sample was stratified as follows: pediatric population (<14 years old), adults from 15 to 64 years old, and adults over 65 years old. The average estimated population of Spain from 1997 to 2017 was 44,323,846 inhabitants; of those, 6,523,950 inhabitants were <14 years old, 30,366,232 inhabitants were 14–65 years old, and 7,683,689 inhabitants were >65 years old.

### 2.3. Data Analysis

In epidemiology, lethality is the proportion of people who die from a specified disease among all individuals diagnosed with the disease over a certain period. It is also known as the case fatality rate (CFR). It is a better measure of the disease’s clinical significance than mortality. It was calculated by dividing the number of deaths associated with *Aspergillus* infection over the defined period (1997–2017) by the number of individuals diagnosed with *Aspergillus* infection during that time. CFR takes values between 0 and 1; the resulting ratio is then multiplied by 100 to yield a percentage (from 0 to 100%). Statistical analysis included a descriptive analysis. Quantitative results are expressed using mean and standard deviation (SD). Qualitative results are expressed in absolute values (n), proportions (n/N), and percentages (%). Bivariate analysis was conducted to study the influence of clinical and epidemiological variables on the dependent variables. Association between categorical variables was assessed by Pearson’s χ^2^ test and odds ratio (OR) estimation. Continuous variables were tested by Student’s *t* test or the Mann–Whitney U test for two groups, depending on their normal or nonnormal distribution. ANOVA (F test) was applied to statistically assess the equality of means between groups. Binary and multinomial logistic regression were used to explain the relationship between one nominal (categorical) dependent variable and one or more independent variables and predict the probabilities of the different possible outcomes. Two-tailed hypothesis tests were used to test statistical significance. Significance was defined al p less than 0.5. The statistical package used was SPSS 26.0 (IBM Corp., Armonk, NY, USA).

### 2.4. Ethics Statement

In this study, medical data from CMBD patients were exploited. These data are organized by the Ministry of Social Services of Health and Equality (MSSSI in Spanish). Researchers working in public and private institutions can request access to databases by completing a questionnaire available on the MSSSI website. In this questionnaire, a signed confidentiality statement is needed. All patient data provided by the CMBD are anonymized and identified by the MSSSI before they are provided to the applicants. Based on this confidentiality statement signed with the MSSSI, researchers cannot provide the data to other researchers, so other researchers must request the data directly from the MSSSI. Study protocol was approved by the Clinical Research Ethics Committee of Investigations Involving Drugs of Cantabria, Spain (CEIMC 2020.353). The procedures described here were carried out in accordance with the ethical standards described in the revised Declaration of Helsinki in 2013.

## 3. Results

Eight thousand eighty (8080) deaths were registered among a total of 32,960 aspergillosis inpatients in the Spanish National Health System between January 1997 and December 2017, which represents 1 death out of every 4 cases of patients hospitalized with aspergillosis (CFR, 24.5%). Figure 1 shows the temporal distribution of the annual number of deaths and CFR. The CFR of inpatients in our study was 25%. The inverse relationship between the two is observed: as the number of cases increased, the CFR gradually decreased over the period, from 25.4 and 27.8% in 1997–1998 to values of 20.6 and 20.8% in 2016 and 2017, respectively, with a crude decrease in the CFR of 5% (and relative 20%).

Table 1 shows the epidemiological and clinical characteristics of the patients who died compared to those who survived. The mean age was higher among patients who died (64.1 ± 17.2 vs. 60.2 ± 19.6; *p* < 0.001). CFRs increased progressively with age and doubled (*p* < 0.001) (Figure 2). Mortality among males was slightly higher than among females [5739/22,383 (26.6%) vs. females 2341/10,577(22.1%), OR = 1.2; 95% CI, 1.1–1.3; *p* < 0.001]. The proportion of deaths among patients with a secondary diagnosis (6584/26,386) was higher than that among patients with a principal diagnosis (1496/6574), 25.0% vs. 22.8% OR = 1.1, 95% CI 1.0–1.2; *p* < 0.001].

Comorbidities were detected in a majority of inpatients with aspergillosis. The main disorders associated in order of frequency were (i) respiratory diseases 26,879 (81.6%), mainly chronic obstructive pulmonary disease (COPD) (14,794; 44.9%) and influenza/pneumonia (10,213; 31%); (ii) neoplasms 11,129 (33.8%), 7222 (21.9%) were hematology neoplasms and 4348 (13.2%) were non-hematology neoplasms; (iii) neutropenia 2668 (8.1%); (iv) HIV infection 944 (2.9%); and v) stem cell transplantation 632 (1.9%) and solid organ transplants 232 (0.7%). A total of 7761 (23.5%) patients had respiratory disease and neoplasms combined. We assessed the CFR of inpatients with aspergillosis according to the main comorbidities (Table 2 and Figure 3). Aspergillosis in patients with respiratory disease and neoplasms together had a CFR (34.2 percent) two times higher than patients with only neoplasms (16.9 percent) or only respiratory comorbidities (23.7 percent) and three times higher than patients with other non-respiratory or non-neoplasm comorbidities (11.8 percent). Influenza and pneumonia occurrence/association significantly increased CFRs in all cases (*p* < 0.001), as well as the measure of disease severity. CFR in aspergillosis in neutropenic patients was 24.4 percent. A total of 1981 inpatients were hospitalized in the critical care unit, and the CFR (94.5 percent) among inpatients in the intensive care unit was significantly higher (*p* < 0.001) (see Table 2). 10.4% (1067/10,213) of influenza/pneumonia patients were admitted to the critical care unit; most of them (95.6%) died (1020/1067).

### Multivariate Logistic Regression Analysis

Multiple logistic regression models predicting lethality statistically (*p* < 0.05) associated with *Aspergillus* infection indicated a relationship among age, sex, different comorbidity groups, and ICU (Table 3).

## 4. Discussion

Aspergillosis could be increasing progressively worldwide [7]. In a previous work of our group in Spain [15], using records of patients hospitalized in the public health system, we showed an increase in the incidence of cases of aspergillosis in the last two decades, and the counts of inpatients with aspergillosis tripled [15]. These same results have also been reported in other areas [16]. This could be due to an increase in classic disorders causing immune suppression associated with aspergillosis, such as leukemia, HSCTs, and SOT [7]. However, other reasons could justify the current epidemiology of aspergillosis, such as an increase in advanced COPD, connective tissue diseases treated with steroids, decompensated cirrhosis or solid tumors, pneumonia caused by the influenza virus, or, more recently, an infection caused by SARS-CoV-2 [17,18,19]. It is possible that a natural increase in life expectancy leads to higher use of chemotherapy, immunosuppressant drugs for transplants, and other clinical conditions in older populations that could contribute to this increased incidence.

Aspergillosis in neutropenic patients is associated with high mortality. Thus, diagnostic algorithms have been proposed to perform early diagnosis and preemptive treatment. An interesting finding of our study was the high frequency of co-infection between influenza virus and Aspergillus. In the cohort analyzed, we detected that one third of the patients admitted had influenza virus. This association has been frequently described in the literature, although only in patients admitted to critical care units [8,20,21,22,23,24]. In our study, only 10% of all of them admitted to the intensive care unit, and most of them died. Some works has shown Aspergillus is present inpatients with influenza virus and COVID-19, from the first days of admission [24,25]. Our work supports the hypothesis that outpatients with these viral infections, may frequently be colonized by *Aspergillus*. Thus, only a small percentage of them, especially the immunosuppressed, will develop more serious clinical symptoms and will be admitted to critical care units. In previous works, the mortality of patients with aspergillosis and influenza virus attended in intensive care units has been between 28–100% [8,20,21,22,23,24]. This wide range could be due to selection bias between the different studies, being higher in invasive pulmonary aspergillosis, use of corticoids, and other factors [23]. Taking into account the high lethality of this coinfection, we believe that *Aspergillus* infection should be sought, not only in patients admitted to the critical care unit but also in risk groups in hospital wards.

Currently, there are few studies that specifically analyze overall mortality over time. In this work, we specifically evaluated the mortality of inpatients with aspergillosis using an indicator called case fatality risk or case fatality ratio, which has a range between 0 and 1 (in percentages from 0 to 100%). Case fatality rates are not constant; they can vary among populations and over time, depending on the interplay between the causative agent of the disease, the environment, and the host, as well as available treatments and quality of patient care [26].

The CFR of inpatients in our study was 25% [27]. Moreover, we detected a progressive decrease in the CFR of close to 5% (decreasing by 20% in relative terms) during the two decades of the study period. The diagnostic and therapeutic improvements that have occurred during the last twenty years may be an explanation for the lower CFR. This has also been found in other studies [11,12,13]. In a similar work, using hospital discharge records of the National Inpatient Sample (NIS) in the USA, Zilberberg and colleagues found a decrease in mortality of 5% in a period similar to our study [16]. The reason for the lower mortality could be due to improved strategies to diagnose and treat fungal diseases [11,12,13].

In our work, we detected a higher CFR in males than in females. We have no clear explanation for this data, although similar results have also been described in other studies [27]. We found a higher CFR among patients older than 45 years. In the previously cited work, the authors did not find differences in CFR according to age [27].

Due to the methodology of our work, we could not establish differences in the CFR among patients with or without invasive aspergillosis or in other clinical settings because the National Health Service, during the time of the study, classified hospitalized patients according to the International Classification of Diseases-9, and the main clinical syndromes were not collected in these records.

Likewise, we were not able to evaluate differences in mortality in neutropenic and nonneutropenic patients as detected by other authors, who found higher mortality rates in the group of nonneutropenic patients compared to neutropenic patients (74–90% vs. 60–66%) [17]. The reason for this difference seems to lie in the diagnostic delay of aspergillosis in the nonneutropenic group due to a lack of clinical suspicion because nonneutropenic patients are less symptomatic than neutropenic patients; notably, there were fewer cases of fever among nonneutropenic patients [28]. Therefore, these patients initially receive antibiotics empirically, and only after failure are antifungal drugs prescribed [29].

On the other hand, we assessed whether comorbidities were associated with different outcomes in inpatients with aspergillosis. Overall, we found that respiratory disorders such as COPD and other non-classic risk factors were the main diseases associated with this mycosis. Recent works have shown the importance of these non-classic risk factors for aspergillosis. Thus, a systematic review of the global impact of *Aspergillus* in COPD found that overall 1.3% of the total patients with COPD are admitted to hospital with invasive aspergillosis [30] and the mortality of these patients could be between 43–71.7% [31,32], being higher than in groups with cancer or hematological disease.

Regarding mortality and different associated comorbidities, we highlight in our work that the factor with the greatest influence on mortality was the presence of influenza and pneumonia alone or associated with tumors, mainly nonhematological tumors [8,20]. In this manner, the rising reports of influenza-associated pulmonary aspergillosis in patients with ARDS, who are otherwise not considered at risk for fungal pneumonia, demand heightened clinical awareness. Tracheobronchitis and *Aspergillus* in respiratory tract samples should prompt suspicion of invasive fungal infection and further work-up. To decrease the burden of influenza-related illness, vaccination is of the utmost importance, specifically in patients with comorbidities.

It should be noted that new factors, such as SARS-CoV-2 infection, were not evaluated in our work given the dates of data collection. In recent studies, aspergillosis must be considered a serious and potentially life-threatening complication in patients with severe COVID-19 receiving immunosuppressive treatment [33]. In a recent study, the 30-day mortality of inpatients in the intensive care unit with proven invasive aspergillosis was twice as high as that in patients with only COVID-19 [34].

The strengths of our work are based on the fact that the CMBD provides information from a network of hospitals that covers more than 99% of the population living in Spain (http://www.msssi.gob.es/, accessed on 1 April 2022); thus, we are confident that this study provides fairly accurate estimates. Due to free access to health services in our country, there are no differences in terms of medical payments or other healthcare-related factors. However, there were several factors that contributed to the limitations of our study. For example, the use of the ICD-9 code in the CMBD has classification limitations with respect to the ICD-10 previously referenced. Most of the cases included in this study were selected following ICD-9-CM (period 1997–2015), which groups all forms of aspergillosis into a single code (117.3), undistinguished. In contrast, ICD-10 discriminates forms of aspergillosis (code B44.0 to B44.9), but these are only cases from 2016 and 2017. Moreover, we did not include cases from private centers or nonhospital cases. Finally, we did not consider deaths from aspergillosis of discharged patients, which could underestimate the mortality and the reason that the details regarding subclassification of aspergillosis could not be established because there was no classification of drugs

Authors should discuss the results and how they can be interpreted from the perspective of previous studies and the working hypotheses. The findings and their implications should be discussed in the broadest context possible. Future research directions may also be highlighted.

## 5. Conclusions

Our study shows that lethality significantly decreased in the last two decades despite the increase in cases. This highlights that patients with solid and/or hematological cancer do not have a much higher mortality rate than the group of patients with pneumonia/influenza alone, these two factors being the ones that cause the highest CFRs. Thus, a high level of suspicion is necessary for the non-oncological group to allow for early diagnosis and timely therapeutic intervention. We also need a greater number of studies that analyze the causes of mortality to decrease it, as well as evaluate the impact of new factors such as COVID-19.

## Figures and Tables

**Figure 1 jof-08-00576-f001:**
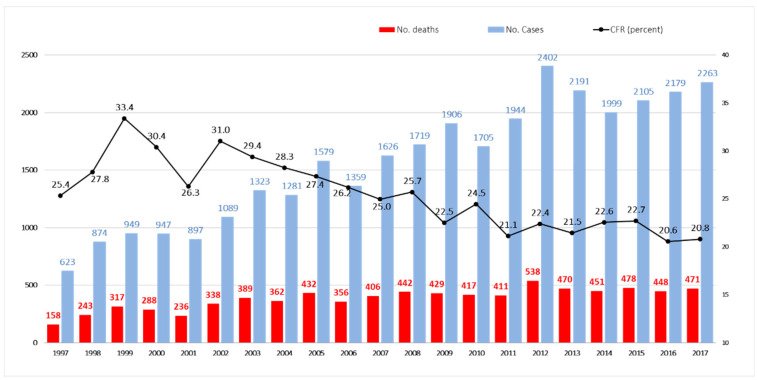
Temporal distribution of annual number of deaths, cases and CFRs (percent) in aspergillosis inpatients in Spain, 1997–2017.

**Figure 2 jof-08-00576-f002:**
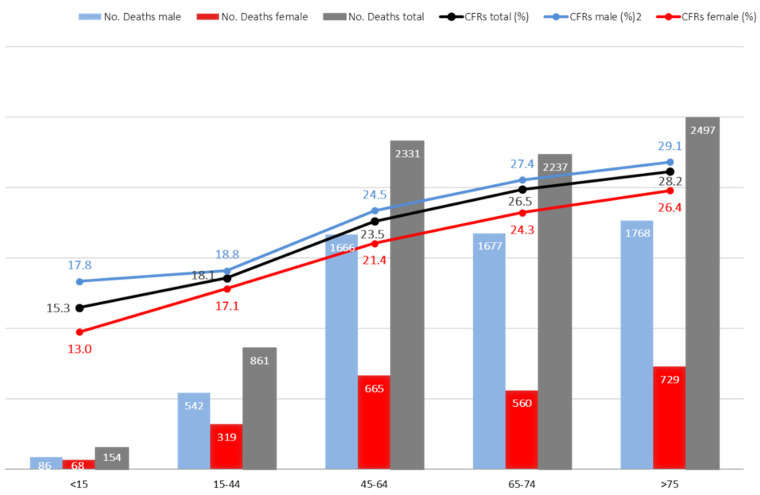
CFRs by age and gender distribution.

**Figure 3 jof-08-00576-f003:**
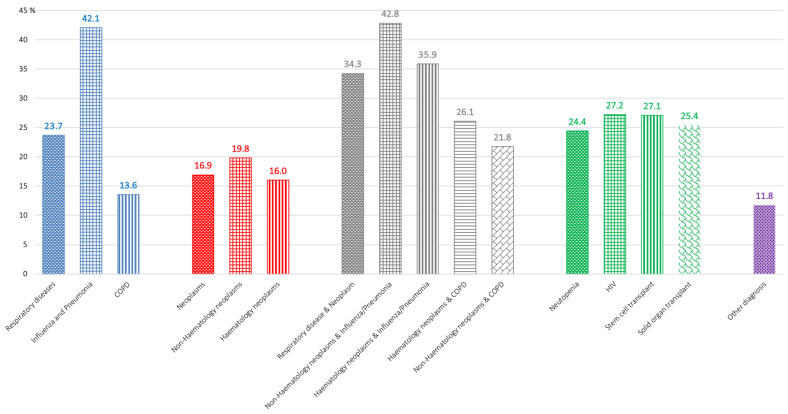
Comorbidities and main Case Fatality Rates (percent).

**Table 1 jof-08-00576-t001:** Main comparative data of *Aspergillus* inpatients in Spain during 1997–2017: Deaths vs. Survivors.

	TOTAL, n (%) N = 32,960 Cases (100%)	Deaths, n (%) N_1_ = 8080 (24.5%)	Survivors, n (%) N_2_ = 24,880 Cases (75.5%)	*p*-Value	OR (95% CI)
**Age**, mean ± SD	61.1 ± 19.1	64.1 ± 17.2	60.2 ± 19.6	<0.001	
<15 years	1009 (3.1)	154 (1.9)	855 (3.4)	<0.001 *	
15–44 years	4753 (14.4)	861 (10.7)	3892 (15.6)		
45–64 years	9921 (30.1)	2331 (28.8)	7590 (30.5)		
65–74 years	8437 (25.6)	2237 (27.7)	6200 (24.9)		
≥75 years	8840 (26.8)	2497 (30.9)	6343 (25.5)		
**Gender**					
Male	22,383 (67.9)	5739 (71.0)	16,643 (66.9)	<0.001 *	1.2 (1.1–1.3)
Female	10,577 (32.1)	2341 (29.0)	8236 (33.1)		
**Diagnosis causing hospitalization**					
Principal diagnosis	6574 (19.9)	1496 (18.5)	5078 (20.4)	<0.001 *	1.1 (1.0–1.2)
Secondary diagnosis	26,386 (80.1)	6584 (81.5)	19,802 (79.6)		
**Process type**					
Medical	21,728 (65.9)	4724 (58.5)	17,004 (68.3)	<0.001 *	0.6 (0.6–0.7)
Surgical	4474 (13.6)	1368 (16.9)	3105 (12.5)		
Unclassified	6758 (20.5)	1987 (24.6)	4771 (19.2)		
**Hospital stays (days)**, mean ± SD	26.8 ± 26.9	32.2 ± 29.4	25.0± 25.8	<0.001 **	

* Pearson’s χ^2^ Test; ** ANOVA; *p* < 0.05 (statistically significant).

**Table 2 jof-08-00576-t002:** CFRs by main comorbidities in aspergillosis inpatients.

		No. Cases	No. Deaths	CFR (Percent)
Solid organ transplant		232	59	25.4
Stem cell transplantation		632	171	27.1
HIV		944	257	27.2
Neutropenia		2668	650	24.4
Respiratory disease		19,118	4533	23.7
Chronic obstructive pulmonary disease (COPD)		9101	1233	13.5
Influenza/ Pneumonia		2999	1262	42.1
COPD & Influenza/Pneumonia		3297	1069	32.5
Other respiratory disease		3721	969	26.0
Neoplasms		3368	570	16.9
Hematology neoplasms		2477	396	16.0
Non-Hematology neoplasms		727	144	19.8
Hematology & Non-Hematology neoplasms		164	30	18.3
Neoplasms & respiratory disease		7761	2658	34.2
Hematology neoplasms &	COPD	472	123	26.1
	Influenza and Pneumonia	2250	807	35.9
	COPD & Influenza and Pneumonia	263	96	36.5
	Other respiratory disease	1319	498	37,7
Non-Hematology neoplasms &	COPD	1144	249	21.8
	Influenza and Pneumonia	789	338	42.8
	COPD & Influenza and Pneumonia	448	156	34.8
	Other respiratory disease	799	302	37.8
Hematology & Non-Hematology neoplasms &	COPD	39	7	17.9
	Influenza and Pneumonia	137	46	33.6
	COPD & Influenza and Pneumonia	30	8	26.7
	Other respiratory disease	71	28	39.4
Other diagnosis		2713	319	11.8
Critical Care Unit *		1981	1872	94.5
	Patient with respiratory disease	1299	1224	94.2
	Patient with neoplasms	39	36	92.3
	Patient with neoplasms & respiratory disease	572	551	96.3
	Patient with other diagnosis	71	61	85.9
Transplant Unit		468	39	8.3
**TOTAL**		32,960	8080	24.5

* Includes Intensive Care Unit, Neonatal Intensive Care Unit, and Pediatric Intensive Care Unit.

**Table 3 jof-08-00576-t003:** Multivariate Logistic Regression Analysis.

Dependent Variable: Exitus Letalis				
Independent Variables	Sig.	Exp(B)	95% CI EXP(B)	
			Lower	Upper
Sex	0.000	1.243	1.167	1.325
Age	0.000	1.882	1.766	2.006
Hematology neoplasms	0.000	1.490	1.380	1.608
Non-Hematology neoplasms	0.000	1.508	1.393	1.633
Influenza/Pneumonia	0.000	2.133	2.012	2.262
COPD	0.000	0.660	0.619	0.705
HIV	0.000	1.357	1.143	1.610
Neutropenia	0.051	0.789	0.705	0.884
ICU	0.000	68.228	55.742	83.510

## Data Availability

Not applicable.

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
