# Peer review of "How Has the Aspergillosis Case Fatality Rate Changed over the Last Two Decades in Spain?"

_jof, 2022, doi:10.3390/jof8060576_

Round 1
Reviewer 1 Report
This study is a descriptive study describing the epidemiologic characteristics of aspergillosis through the national health system in Spain. I think this paper is meaningful in that it reviewed the epidemiology of aspergillosis in a country.
I am curious as to whether or not you have received English correction. If you have not received it, we recommend that you take the correction again.
Other details are as follows.
Line 55-57 is equivalent to line 50-52. It is recommended to remove duplicate items.
In line 66, epidemiologic studies related to aspergillosis are reported worldwide, but studies in Spain may be insufficient. I recommend describing previous epidemiological studies and add relevant references.
In line 83, it is better to describe the full name, CBMD.
In line 180-188, it may be meaningful to obtain the CFR for the entire aspergillosis, but aspergillosis is a broad disease entity. It is necessary to obtain CFR by dividing by IPA, CNPA, aspergilloma, etc. It is thought that it can be estimated through the list of antifungal drugs used.
In line 211, numerical quantification is required for aspergillosis in neutropenic patients. In addition, since this study did not only cover neutropenic patients, data descriptions from previous studies of the entire epidemiology are required.
In line 225-228, it would be helpful to state the CFR values from previous studies.
In line 251, I recommend to describe the previous study regarding the relevance between respiratory disease and aspergillosis, and the interpretation of this study.
As a limitation of this study, it should be emphasized that only aspergillosis is described in this study, not subtype sush as IPA, CNPA, or aspergilloma, etc.
Author Response
Response to Reviewer 1
This study is a descriptive study describing the epidemiologic characteristics of aspergillosis through the national health system in Spain. I think this paper is meaningful in that it reviewed the epidemiology of aspergillosis in a country.
I am curious as to whether or not you have received English correction. If you have not received it, we recommend that you take the correction again.
RESPONSE: Our paper was edited for proper English language and certified by the American Journal Experts (AJE) and could be verified on the AJE website using the verification code 2A63-200B-84E1-189D-11CA. However, we have redrafted some paragraph for to avoid plagiarism.
Other details are as follows.
Line 55-57 is equivalent to line 50-52. It is recommended to remove duplicate items.
RESPONSE: Done. Thanks.
In line 66, epidemiologic studies related to aspergillosis are reported worldwide, but studies in Spain may be insufficient. I recommend describing previous epidemiological studies and add relevant references.
As suggested the reviewer, we have included two relevant references of the epidemiology of Aspergillosis in Spain.
In line 83, it is better to describe the full name, CMBD.
RESPONSE: Done.
In line 180-188, it may be meaningful to obtain the CFR for the entire aspergillosis, but aspergillosis is a broad disease entity. It is necessary to obtain CFR by dividing by IPA, CNPA, aspergilloma, etc. It is thought that it can be estimated through the list of antifungal drugs used.
RESPONSE: These data could not be estimated because ICD-9-CM groups all forms of aspergillosis into a single code (117.3), undistinguished (cases from 1997 to 2015). In contrast, ICD-10 discriminates forms of aspergillus (code B44.0 to B44.9), but these are only cases from 2016 and 2017. In addition, CMBD does not record data on drugs.
In line 211, numerical quantification is required for aspergillosis in neutropenic patients. In addition, since this study did not only cover neutropenic patients, data descriptions from previous studies of te entire epidemiology are required.
RESPONSE: We have quantified aspergillosis in neutropenic patients (lines 177-178, 187, and table 2). Paragraph is introduced in the discussion.
In line 225-228, it would be helpful to state the CFR values from previous studies.
We agree with the reviewer and we have included in the line 225 the reference 18 about the CFR in other studies.
In line 251, I recommend to describe the previous study regarding the relevance between respiratory disease and aspergillosis, and the interpretation of this study.
We do not understand exactly what the reviewer is referring to.
As a limitation of this study, it should be emphasized that only aspergillosis is described in this study, not subtype such as IPA, CNPA, or aspergilloma, etc.
RESPONSE: Done. We have added a sentence explaining this aspect in the study limitations (in line 279-282).
Reviewer 2 Report
This manuscript presents very important information that will prove useful in the review of the management of aspergillosis cases as it provides a comprehensive review of inpatient aspergillosis cases for the last 20 years.
Comments to the authors
Line 24: …due to this fungus is not… Aspergillosis is not a fungus.
Line 28-29: Does the 71% (cases in men) relate to the total number of inpatients or the proportion of death? Please clarify
Line 29-31: I would suggest that you state the percentage of increased death and case fatality rate in the abstract. The statement as it is needs to be read with the statistic in mind.
Line 33-34: I am not comfortable with the comparison made between patients with cancer and those with pneumonia/influenza here. No results for the cancer group were mentioned in the abstract before this comparison was made. Also, the comparison was made in the conclusion, which is a bit strange. I would feel more comfortable if you can show the results for the cancer group before comparing it to other patient groups or just take it out from the abstract. This is a very minor comment about the writing style and not about the content of the work stated here.
Line 105-106: Is there a reason why the authors terminated the paediatric age group at 14 and not 18, and why 15 - 18 were treated as adults? Is this a standard practice or an arbitrary cut-off?
Line 105-108: <14 means less than 14, that is 1-13. Are the authors aware of this? If the intention is to include age 14 then ≤14 would be the right symbol to use.
Line 155: Figure 1. The authors should supply a higher resolution figure. Can the authors use a different colour/font for the numbers on the figure? It was quite hard for me to see all the details.
Line 160: Can the authors please explain the rationale behind the age groups in Table 1? Are these distinct age groups (biologically) or an arbitrary grouping?
Line 170: Figure 2: Is there a reason why the authors had 10 years sliding scale for age distribution in figure 2? What is the point of this analysis seeing you have used a different age scale previously (0-14, 15-44, 45-64, 65-74 and ≥75) in Table 1? Can the authors repeat the analysis in Figure 2 using the same age groups in Table 1? Try to include a third group without the gender categorisation.
Line 172-190: This section is highly interesting. Can the authors confirm if the comorbidities reported here were primary or secondary diagnoses? It will be interesting to see a report of the CFR in cases where aspergillosis was the primary diagnosis vs secondary diagnosis.
Table 2: Can the authors please perform a binomial logistic regression on the data presented in table 2 so that the reader can appreciate the contribution of each comorbidity to the reported CFR? There might be other statistical analyses you can think of instead of what I have proposed but there is a need to understand what comorbidity contributes the most to the CFR while taking the effect sizes into account.
Author Response
Response to Reviewer 2
This manuscript presents very important information that will prove useful in the review of the management of aspergillosis cases as it provides a comprehensive review of inpatient aspergillosis cases for the last 20 years.
Comments to the authors
Line 24: …due to this fungus is not… Aspergillosis is not a fungus.
As suggested the reviewer, we have changed Aspergillosis by Aspergillus.
Line 28-29: Does the 71% (cases in men) relate to the total number of inpatients or the proportion of death? Please clarify
We have changed this sentence” Among a total of 32,960 aspergillosis inpatients, 24.5% deaths were registered, and 71% of the inpatients were men” by” Among a total of 32,960 aspergillosis inpatients, 24.5% deaths were registered, being 71% of the patients died were men”
Line 29-31: I would suggest that you state the percentage of increased death and case fatality rate in the abstract. The statement as it is needs to be read with the statistic in mind.
We have included as suggested the reviewer: Case fatality rate (CFR) gradually decreased over the period, from 25.4 and 27.8% in 1997-1998 to values of 20.6 and 20.8% in 2016 and 2017, a crude decrease in the CFR of 5% (and relative 20%).
Line 33-34: I am not comfortable with the comparison made between patients with cancer and those with pneumonia/influenza here. No results for the cancer group were mentioned in the abstract before this comparison was made. Also, the comparison was made in the conclusion, which is a bit strange. I would feel more comfortable if you can show the results for the cancer group before comparing it to other patient groups or just take it out from the abstract. This is a very minor comment about the writing style and not about the content of the work stated here.
As the reviewer refers, the writing style is variable.
Line 105-106: Is there a reason why the authors terminated the paediatric age group at 14 and not 18, and why 15 - 18 were treated as adults? Is this a standard practice or an arbitrary cut-off?
As comments the reviewer, we selected 15 y because in Spain, the children <15 years of age are usually treated by pediatricians, while those over ≥15 years of age are treated as adults.
Line 105-108: <14 means less than 14, that is 1-13. Are the authors aware of this? If the intention is to include age 14 then ≤14 would be the right symbol to use.
We have modified the text as suggested the reviewer.
Line 155: Figure 1. The authors should supply a higher resolution figure. Can the authors use a different colour/font for the numbers on the figure? It was quite hard for me to see all the details.
RESPONSE: Done.
Line 160: Can the authors please explain the rationale behind the age groups in Table 1? Are these distinct age groups (biologically) or an arbitrary grouping?
Line 170: Figure 2: Is there a reason why the authors had 10 years sliding scale for age distribution in figure 2? What is the point of this analysis seeing you have used a different age scale previously (0-14, 15-44, 45-64, 65-74 and ≥75) in Table 1? Can the authors repeat the analysis in Figure 2 using the same age groups in Table 1? Try to include a third group without the gender categorization.
RESPONSE: Done. We have used the same scale for the age distribution in table and figure. In addition, we have included a third group corresponding to total data.
Line 172-190: This section is highly interesting. Can the authors confirm if the comorbidities reported here were primary or secondary diagnoses? It will be interesting to see a report of the CFR in cases where aspergillosis was the primary diagnosis vs secondary diagnosis.
RESPONDE: The comorbidities reported were total data, not differentiated between primary and secondary diagnoses. There is no significant difference between the primary and secondary main diagnosis. Only the presence of comorbidities as indicated by multiple regression.
Table 2: Can the authors please perform a binomial logistic regression on the data presented in table 2 so that the reader can appreciate the contribution of each comorbidity to the reported CFR? There might be other statistical analyses you can think of instead of what I have proposed but there is a need to understand what comorbidity contributes the most to the CFR while taking the effect sizes into account.
RESPONSE: Done. Table 3 is attached.

Reviewer 3 Report
Gonzalez-Garcia et al provide observational data on aspergillosis case fatality rate in inpatients in Spain during a 20-year period. While the results nicely show a decrease in CFR in aspergillosis patients, several major issues arise in the introduction, results and discussion.
Introduction
- Line 42-46: it is a bit strange to put allergic bronchopulmonary aspergillosis in between CPA and IPA, as the spectrum goes from a "hyper" immune response to Aspergillus (ABPA) to an "insufficient" immune response (colonization, then CPA and at the extreme IPA). See the paper by Kosmidis et al that is cited here for more explanation.
- Line 47-50: wrong reference for this sentence.
- Line 50-52: wrong reference, this reference only handles aspergillosis in lung transplant patients and says nothing about stem cell transplants.
- Line 53-54: referring actual studies identifying influenza and COVID-19 as independent risk factors for IPA would be better (e.g. Schauwvlieghe Lancet Resp Med 2018 for influenza and one of the three very large observational studies for COVID-19: Janssen Emerging Infectious Diseases 2021, Prattes Clinical Microbiol Infect and/or Gangneux Lancet Resp Med 2021).
- Line 55-57: same sentence as line 50-52, please remove.
- Line 57-61: this sentence is almost literally copied from the paper (referred in the middle of the sentence, while the second half of the sentence is largely copied from this paper as well). The only difference the authors made to the sentence is the introduction of the words "similar to", which make the meaning of this sentence confusing. This sentence has to be rewritten (right now this is almost plagiarism) and the meaning of the sentence has to be checked.
- Line 61-65: this is plagiarism. These sentences have been exactly copied from Latgé and Chamilos, Clin Microbiol Rev 2020, with the exception for breaking the copied sentence in two. The references here are taken from the Latgé & Chamilos paper, while a reference to this paper is missing in this section.
Conclusion: the introduction needs to be completely rewritten with proper references and without plagiarism.
Methods
- Please state whether a one-sided or two-sided hypothesis was used.
Results
- Higher quality figures are necessary, in Figure 1 the numbers are almost not readable.
- CFR is often mistakenly written as CRF, please adjust. Also, the abbreviation CFR is explained multiple times, please adjust.
- It would be interesting to see what the CFR was for influenza patients admitted to the critical care unit. If this is >90% as seen in the general respiratory disease population in this study, this would be extremely high compared to other pbservational studies (e.g. +/- 50% in Schauwvlieghe et al).
- Also, it would be very interesting to see how many patients with influenza and aspergillosis (IAPA) required admission to a critical care unit, and how many did not. This would be a novelty, as right now in literature we only have a good view on the incidence of aspergillosis in critically ill influenza patients. Please include these data in the tables and body of the text.
- The data on the multivariate logistic regression analysis is missing. Only stating that these analyses indicated a relationship among age, sex, and differenct comorbidities is not enough: all results should be provided in this alinea, along with tables so we may interpret the data...
Discussion
- The authors (again) explain what CFR is in line 213-224. Apparently, line 218-224 is almost exactly copy/pasted from the Britannica-website (https://www.britannica.com/science/case-fatality-rate). Not only is this part not suited for a discussion, it is also (again) plagiarism...
- Line 225-226: this is a strange sentence, do the authors mean that in ref 11 and 17 the CFR was lower? In ref 11 and 17, CFR is not used so this statement is wrong anyway.
- The authors should elaborate more on what diagnostic and therapeutic improvements happened during the last twenty years (e.g. galactomannan testing), which may explain the lower CFR.
Author Response
Response to Reviewer 3
Gonzalez-Garcia et al provide observational data on aspergillosis case fatality rate in inpatients in Spain during a 20-year period. While the results nicely show a decrease in CFR in aspergillosis patients, several major issues arise in the introduction, results and discussion.
Introduction
Line 42-46: it is a bit strange to put allergic bronchopulmonary aspergillosis in between CPA and IPA, as the spectrum goes from a "hyper" immune response to Aspergillus (ABPA) to an "insufficient" immune response (colonization, then CPA and at the extreme IPA). See the paper by Kosmidis et al that is cited here for more explanation.
We agree with the reviewer and we have changed this paragraph following the reference of Kosmidis et al.
Line 47-50: wrong reference for this sentence.
We have modified the reference.
Line 50-52: wrong reference, this reference only handles aspergillosis in lung transplant patients and says nothing about stem cell transplants.
As suggested, we have changed this reference
Line 53-54: referring actual studies identifying influenza and COVID-19 as independent risk factors for IPA would be better (e.g. Schauwvlieghe Lancet Resp Med 2018 for influenza and one of the three very large observational studies for COVID-19: Janssen Emerging Infectious Diseases 2021, Prattes Clinical Microbiol Infect and/or Gangneux Lancet Resp Med 2021).
We have modified the reference
Line 55-57: same sentence as line 50-52, please remove.
We have modified the sentence
Line 57-61: this sentence is almost literally copied from the paper (referred in the middle of the sentence, while the second half of the sentence is largely copied from this paper as well). The only difference the authors made to the sentence is the introduction of the words "similar to", which make the meaning of this sentence confusing. This sentence has to be rewritten (right now this is almost plagiarism) and the meaning of the sentence has to be checked.
We have modified the sentence
Line 61-65: this is plagiarism. These sentences have been exactly copied from Latgé and Chamilos, Clin Microbiol Rev 2020, with the exception for breaking the copied sentence in two. The references here are taken from the Latgé & Chamilos paper, while a reference to this paper is missing in this section.
RESPONSE: PENDIENTE MON.
We have modified the sentence.
Conclusion: the introduction needs to be completely rewritten with proper references and without plagiarism.
We have modified the introduction.
Methods
Please state whether a one-sided or two-sided hypothesis was used.
RESPONSE: The purpose of this retrospective longitudinal descriptive study was to collect information and to generate future research hypotheses. It was not designed to verify hypotheses.
Results
Higher quality figures are necessary, in Figure 1 the numbers are almost not readable.
RESPONSE: Done.
CFR is often mistakenly written as CRF, please adjust. Also, the abbreviation CFR is explained multiple times, please adjust.
RESPONSE: Done.
It would be interesting to see what the CFR was for influenza patients admitted to the critical care unit. If this is >90% as seen in the general respiratory disease population in this study, this would be extremely high compared to other pbservational studies (e.g. +/- 50% in Schauwvlieghe et al).
RESPONSE: We have added a sentence in lines 189-190: “10.4% (1,067/10,213) of influenza/pneumonia patients were admitted to the critical care unit; most of them (95.6%) died (1,020/1,067)”.
Also, it would be very interesting to see how many patients with influenza and aspergillosis (IAPA) required admission to a critical care unit, and how many did not. This would be a novelty, as right now in literature we only have a good view on the incidence of aspergillosis in critically ill influenza patients. Please include these data in the tables and body of the text.
RESPONSE: Done in the previous point.
The data on the multivariate logistic regression analysis is missing. Only stating that these analyses indicated a relationship among age, sex, and differenct comorbidities is not enough: all results should be provided in this alinea, along with tables so we may interpret the data...
RESPONSE: Done. Table 3 is attached.
Discussion
The authors (again) explain what CFR is in line 213-224. Apparently, line 218-224 is almost exactly copy/pasted from the Britannica-website (https://www.britannica.com/science/case-fatality-rate). Not only is this part not suited for a discussion, it is also (again) plagiarism...
We have modified the sentence.
Line 225-226: this is a strange sentence, do the authors mean that in ref 11 and 17 the CFR was lower? In ref 11 and 17, CFR is not used so this statement is wrong anyway.
We have modified the sentence.
The authors should elaborate more on what diagnostic and therapeutic improvements happened during the last twenty years (e.g., galactomannan testing), which may explain the lower CFR.
We have modified the sentence.

Round 2
Reviewer 1 Report
- In 71, In Spain, two previous work have showed an increasing in the counts of Aspergillosis.
--> There still seems to be a problem with English grammar. - In line 251, I recommend to describe the previous study regarding the relevance between respiratory disease and aspergillosis, and the interpretation of this study.
--> This meant that it was necessary to review the literature on the relationship between previous respiratory diseases and aspergillosis and compare it with the results of this study. - In the limitation, I recommend to describe that the reason that the details regarding subclassification of aspergillosis could not be established because there was no classification of drugs.
Author Response
Reviewer 1 Round 2
- In 71, In Spain, two previous work have showed an increasing in the counts of Aspergillosis.
I don't understand the comment
- There still seems to be a problem with English grammar.
Some parts have been modified for better understanding.
- In line 251, I recommend to describe the previous study regarding the relevance between respiratory disease and aspergillosis, and the interpretation of this study.
--> This meant that it was necessary to review the literature on the relationship between previous respiratory diseases and aspergillosis and compare it with the results of this study.
I don't understand the comment, Line 251 is missing
- In the limitation, I recommend to describe that the reason that the details regarding subclassification of aspergillosis could not be established because there was no classification of drugs.
Comment is included

Reviewer 3 Report
Most issues have been addressed accordingly by the authors.
- The authors still need to say whether they performed statistics using a one-sided or two-sided hypothesis to perform p-value calculation. Using statistics implies that one tests a hypothesis (e.g. whether there is a difference in mean age in aspergillosis patients who died or survived), thus the authors must state whether they used a one-sided or two-sided hypothesis to calculate their p-values. Most likely, the authors have used a two-sided hypothesis but this must be stated in the manuscript...
- Do the authors have an explanation for the fact that only 10% of influenza/pneumonia patients were admitted to critical care unit, while the rest of the literature has mainly described influenza-associated aspergillosis in critically ill patients? Is underregistration of critical care unit admission a possible explanation? Or is registration of mono-infection with aspergillosis as "pneumonia" a possibility in the tool? If so, what was the proportion of influenza patients (and not influenza and pneumonia) with aspergillosis admitted to critical care unit? Do the authors know how many influenza patients had other host factors for aspergillosis as well?
- The multivariate logistic regression analysis table is merely a table copied from a statistics program. This should be summarized as a table (ideally with a forest plot) table containing the independent variable name, the odds ratio with confidence interval and the p-value. Mentioning the constant, the beta, standard error, Wald, gl is not needed. "Respiratory" and "Oncology" need more clarification.
The English language needs review in the newly added/adjusted parts.
Author Response
Reviewer 3 round 2
- Most issues have been addressed accordingly by the authors.
Thanks for the feedback
- The authors still need to say whether they performed statistics using a one-sided or two-sided hypothesis to perform p-value calculation. Using statistics implies that one tests a hypothesis (e.g. whether there is a difference in mean age in aspergillosis patients who died or survived), thus the authors must state whether they used a one-sided or two-sided hypothesis to calculate their p-values. Most likely, the authors have used a two-sided hypothesis but this must be stated in the manuscript...
RESPONSE: We used a two-tailed hypothesis test. SPSS calculates the p-value for a bilateral (or two-tailed) contrast in the Chi-square test; p-value (bilateral asymptotic significance).
- Do the authors have an explanation for the fact that only 10% of influenza/pneumonia patients were admitted to critical care unit, while the rest of the literature has mainly described influenza-associated aspergillosis in critically ill patients? Is underregistration of critical care unit admission a possible explanation? Or is registration of mono-infection with aspergillosis as "pneumonia" a possibility in the tool? If so, what was the proportion of influenza patients (and not influenza and pneumonia) with aspergillosis admitted to critical care unit? Do the authors know how many influenza patients had other host factors for aspergillosis as well?
RESPONSE: Whether Aspergillus co-infection increases the risk of admission to the ICU is a good question for another paper but not this one. The text has been modified and completed for a better understanding.
- The multivariate logistic regression analysis table is merely a table copied from a statistics program. This should be summarized as a table (ideally with a forest plot) table containing the independent variable name, the odds ratio with confidence interval and the p-value. Mentioning the constant, the beta, standard error, Wald, gl is not needed. "Respiratory" and "Oncology" need more clarification.
RESPONSE: Table 3 has been summarized as indicated by the reviewer a forest graph has been added. "Respiratory" and "Oncology" has been clarified.
- The English language needs review in the newly added/adjusted parts.
Some parts have been modified for better understanding.
